# Soybean Bioactive Peptide Supplementation Affects the Intestinal Immune Antioxidant Function, Microbial Diversity, and Reproductive Organ Development in Roosters

**DOI:** 10.3390/ani14131954

**Published:** 2024-07-02

**Authors:** Yimeng Wei, Xiyu Zhao, Tao Xu, Zhenyan Liu, Yalan Zuo, Mingxue Zhang, Yao Zhang, Huadong Yin

**Affiliations:** 1Key Laboratory of Livestock and Poultry Multi-Omics, Ministry of Agriculture and Rural Affairs, College of Animal Science and Technology, Sichuan Agricultural University, Chengdu 611130, China; w15213090968@outlook.com (Y.W.); zhaoxiyu@stu.sicau.edu.cn (X.Z.); 19141397285@163.com (T.X.); lzy13684281345@163.com (Z.L.); zuoyalan@stu.sicau.edu.cn (Y.Z.); zhangmingxue@stu.sicau.edu.cn (M.Z.); zhangyao@sicau.edu.cn (Y.Z.); 2Farm Animal Genetic Resources Exploration and Innovation Key Laboratory of Sichuan Province, Sichuan Agricultural University, Chengdu 611130, China

**Keywords:** soybean bioactive peptides, rooster, immune, antioxidant, reproductive performance

## Abstract

**Simple Summary:**

This study explored the impact of different levels of soybean bioactive peptide feed additives (0%, 0.15%, 0.30%, 0.45%, and 0.60%) on the health and reproductive performance of roosters. The results showed that soybean bioactive peptide supplementation significantly improved growth rate, feed efficiency, reproductive organ development, and semen quality. Additionally, soybean bioactive peptides enhanced immune and antioxidant functions, improved gut health by maintaining the small intestine’s structure and barrier, and increased the diversity of beneficial gut microbes while reducing cell death in the intestinal lining. The benefits of soybean bioactive peptides were most pronounced at a 0.45% concentration. Therefore, incorporating 0.45% soybean bioactive peptides into rooster diets is optimal for enhancing their overall health and reproductive performance.

**Abstract:**

Soybean is an important source of high-quality vegetable protein with various health-improving properties, and its main bioactive substances are small peptides produced by in vitro enzymatic hydrolytic processes. In traditional layer breeding, the nutritional health of roosters is frequently neglected, ultimately affecting the quality and quantity of offspring. This study investigated the effects of various quantities (0%, 0.15%, 0.30%, 0.45%, and 0.60%) of soybean bioactive peptide (SBP) feed additives on immunological and antioxidant functions, gut health, and reproductive performance of roosters. SBP supplementation significantly improved male growth and reproductive performance, including growth rate, feed conversion ratio, reproductive organ development, and semen quality. SBP also increased immune and antioxidant levels, boosted the integrity of the small intestinal physiological structure and barrier function, and diversity of cecal microbes, and decreased the apoptotic ratio of small intestinal epithelial cells. The effects of SBP on various functions of males showed a quadratic trend, with the optimal concentration determined to be 0.45%.

## 1. Introduction

In modern poultry production, maintaining optimal health and performance of birds is of paramount importance for sustainable and profitable operations. With the increasing demand for antibiotic-free poultry products and growing concerns about antimicrobial resistance, there is a pressing need for effective alternatives to antibiotics that can support poultry health and welfare [1]. In this context, the exploration of natural feed additives with bioactive properties has garnered considerable attention [2].

SBPs, derived from soy protein through enzymatic hydrolysis or fermentation processes [3,4,5], have emerged as a promising candidate for enhancing poultry health and production [6]. These peptides possess diverse biological activities that can positively influence various aspects of poultry physiology, particularly in intensive farming systems [7,8]. Firstly, SBPs are end products of protein digestion that are more easily absorbed by animal intestine [9,10]. By supplementing poultry feed with these peptides, producers can ensure that birds receive essential nutrients necessary for growth, development, and reproductive performance, and can also enhance intestinal structure and stimulate intestinal tract development [11]. Previous report showed that SBP could improve jejunal villus height and ileal apparent nutrient digestibility of broilers [12]. In addition, the anti-inflammatory, antioxidant, and immunomodulatory functions of SBP can maximize the guarantee and heighten the growth performance and reproductive capacity of livestock and poultry. It is reported that SBP could alleviate intestinal morphological damage and reduce the inflammatory response to coccidioidal challenge [13]. Furthermore, maintaining a balanced gut microbiota is essential for poultry health and performance [14]. SBPs possess antimicrobial activity against a wide range of pathogens, which can help to promote a healthy gut microbiota and reduce the risk of gastrointestinal infections in poultry [15].

During the breeding process of chicken production, the nutritional health of hens has often been emphasized, while that of roosters has been neglected or ignored, eventually leading to poor sire quality and negatively affecting the quantity and quality of offspring. The aforementioned beneficial health properties of soybean-derived SBP make it a good candidate as a dietary supplement for improving the health of breeding roosters. Therefore, in this study, we added different graded concentrations of SBP into the birds’ feed to determine the effects on the following: structural integrity, developmental status, and immune-antioxidant function of each segment of the small intestine; cecal microbial diversity; status of reproductive organ development; and semen quality. Understanding the mechanisms underlying the bioactivity of soybean peptides in poultry will not only contribute to optimizing poultry production systems but also address current challenges related to antibiotic use and antimicrobial resistance in the poultry industry.

## 2. Materials and Methods

### 2.1. Animals, Diets, and Experimental Design

Nine hundred one-day-old Tianfu green-shell chicks with similar body weights (33–34 g) were selected for study. The treatments were randomly distributed into 5 groups, with 6 repetitions of 30 animals each. The control feed group (CONF) was fed a basal commercial diet (New Hope Group Co., Ltd., Beijing, China) and the diets of the four SBP feed groups (SBPF1–SBPF4) were supplemented with different concentrations of SBP (0%, 0.15%, 0.30%, 0.45%, and 0.60%). The roosters were fed for 30 weeks until they were sexually mature and eligible for mating. The bioactive peptide compounds used in this trial were provided by Mytech Company (SP900 Type C, Chengdu, China), and SBP was produced by combining liquid-phase enzymatic hydrolysis with peeled soybean protein concentrate and soybean meal (the molecular weight distribution is presented in Table A1). Additionally, the SBP product is in powdered form. The acid-soluble protein content was determined using the trichloroacetic acid (TCA) precipitation method, ensuring a minimum peptide content of 30.0%, and the specific molecular weight distribution is presented in Table A1. Moreover, the product’s antioxidant capacity ranges from 25 to 30 mmol/kg. Experimental chickens were feeding in Sichuan Agricultural University (Ya’an, China). The feeding and treatment of chicks followed the guidance from the Animal Welfare Committee of Sichuan Agricultural University (Approval number: 2023102023).

### 2.2. Animal Management

The breeding environment for roosters utilized a closed henhouse with longitudinal ventilation, controlled by a central environmental control system. Each rooster was housed individually in a cage (40 cm × 22 cm × 45 cm), with feeding management conducted according to the “Feeding Management Measures for Tianfu Layer Breeding Chickens”. During the experiment, roosters had ad libitum access to feed, and were fed four times daily, with the basal diet formulation detailed in Table A2. Water was provided through nipple drinkers, and the lighting schedule followed the guidelines outlined in the “Feeding Management Measures for Tianfu Layer Breeding Chickens” (see Table A3).

### 2.3. Measurement and Calculation of Growth Performance

The growth performance indices, including average daily gain (ADG), average daily feed intake (ADFI), and feed-to-gain ratio (F/G), were calculated weekly for all birds. Additionally, body weights (BWs), tibia lengths, and feed consumption were recorded. All measurements followed the guidelines outlined in “Nomenclature and Statistical Methods for Measurement of Poultry Performance” (NY/T823-2020).

### 2.4. Serum Collection and Indicator Determination

At the end of the feeding experiment, six chickens were randomly selected from each group, one for each replicate, and 5 mL of venous blood was collected with a non-anticoagulating blood collection tube. The blood was left in a water bath at 37 °C until the serum was separated, and centrifuged at 3000× *g* r/min for 15 min. Serum was then collected for subsequent indicator determination.

Serum-related indices were assessed in terms of immune-related immunoglobulins (IgA, IgM and IgG), inflammation-related interleukins (IL-1β, IL-6, IL-8, and IL-12), antioxidant-related superoxide dismutase (SOD), total antioxidant capacity (T-AOC), glutathione peroxidase (GSH-Px), and malondialdehyde (MDA) by ELISA kits (MeiMian, Yancheng, China), in strict accordance with the manufacturer’s instructions.

### 2.5. Intestinal and Testis TissueSample Collection

After the feeding experiment, three chickens in each group were randomly selected for euthanasia by carotid bloodletting. Subsequently, the duodenum, jejunum and ileum were separated, and food residues in the intestinal lumen were removed. Each intestinal segment was split into two sections; one part was used for RNA extraction, while the other part was fixed in 4% paraformaldehyde for paraffin section preparation. In addition, the cecal contents of each rooster were collected for 16S rRNA sequencing.

### 2.6. Intestinal and Testis Histomorphology

The intestinal and testis tissues were paraffin-sectioned and stained with hematoxylin/eosin (HE) after being fixed with 4% paraformaldehyde solution. The morphological features of each sample were then observed under a microscope (Nikon, Tokyo, Japan) and images were captured. Finally, Image-Pro Plus (Media Cybernetics, Rockville, MD, USA) was used to measure the villus height and crypt depth of the small intestine, the diameter of the seminiferous tubule, and the thickness of the seminiferous epithelium of the testis.

### 2.7. Immunofluorescence

Intestinal and testis tissue samples were treated as follows for immunofluorescence. Firstly, xylene and gradient ethanol were used for deparaffinization, followed by proteinase K for antigen repair, and then stained using a TDT-mediated dUTP Nick-end Labeling (TUNEL) detection kit (Elabscience, Wuhan, China). Secondly, 4′,6-diamidino-2-phenylindole (DAPI, Beyotime, Beijing, China) was used to stain the nuclei. The prepared sections were observed and imaged under an inverted fluorescence microscope (Olympus, Tokyo, Japan). Thirdly, the tissue sections were blocked with bovine serum albumin after antigen retrieval. Fourthly, the sections were incubated with diluted Tight junction protein 1 (known as ZO-1) antibody (ABclonal, Wuhan, China) for 30 min at 37 °C. Following washing, the samples were stained with a Cy3 goat anti-rabbit secondary antibody (ABclonal, Beijing, China). Finally, images were taken using a fluorescence inverted microscope (Olympus, Tokyo, Japan).

### 2.8. Detection of Intestine-Related Gene Expression

After low-temperature grinding, total RNA of intestinal samples was extracted by the trizol (Takara, Dalian, China) method. RNA samples were reverse-transcribed into cDNA samples using a PrimeScript^®^ RT kit (Takara, Dalian, China), according to the manufacturer’s instructions. Subsequently, mRNA expression levels of tight junction proteins, immune factors, and antioxidant factors for the duodenum, jejunum, and ileum were detected by quantitative PCR. Relative gene expression was calculated by the 2^−△△CT^ method. The primers are presented in Table A4.

### 2.9. Semen Quality Determination

Six roosters in each group were randomly selected for the collection of semen over three consecutive days by abdominal and back massage, and the ejaculate volume was recorded. Subsequently, sperm density, sperm motility, and abnormality rate were determined with eosin−nigrosine and Giemsa stain (Solarbio, Beijing, China). In addition, semen pH was measured with a pH meter (INESA, Shanghai, China).

### 2.10. Cecal Microbial Sequencing Analysis

Firstly, total DNA was extracted from cecal content samples using the cetyltrimethylammonium bromide method. Secondly, the V3−V4 hypervariable regions of the 16S rRNA gene were amplified by PCR using primers 515F and 806R. After purification and quantification, PCR products were sequenced on an Illumina NovaSeq platform. Microbiome phenotypes were predicted with BugBase “https://bugbase.cs.umn.edu/ (accessed on 18 January 2024)”.

### 2.11. Statistical Analysis

Statistical analysis and graph plotting were performed using SPSS v.19.0 (IBM, Armonk, NY, USA). Data analysis methods included one-way ANOVA, Duncan’s multiple comparison analysis, and linear- and quadratic-effects analysis. All data are presented as mean and standard error (SEM). Statistical significance was set at α = 0.05. Gut microbiota was analyzed using nonmetric multidimensional scaling (NMDS) and principal coordinate analysis (PCoA).

## 3. Results

### 3.1. SBP Supplementation Improves the Growth Performance of Roosters

The growth performance of roosters in each group is shown in Table 1. By the sixth week, the mean body weights of birds in the four SBPF groups (SBP feed groups 1–4) began to significantly exceed that of the CONF group (*p* < 0.05). By the 30th week, the mean body weights of the SBPF2, SBPF3, and SBPF4 groups were more than 150 g higher than that of the CONF group (*p* < 0.05), with the most pronounced effect seen in the SBPF3 group. Moreover, both ADG and BW showed similar trends, first increasing and then decreasing with increasing amounts of added SBP, consistent with a quadratic trend (P_Q_ < 0.05). Dietary supplementation with different concentrations of SBP had no significant effect on the roosters’ feed intake (*p* > 0.05), although F/G was significantly reduced (*p* < 0.05). Furthermore, SBP improved ADFI in a quadratic manner and decreased F/G in a linear manner (P_L_ < 0.05). At the sixth week, higher concentrations of SBP significantly enhanced rooster tibia length (*p* < 0.05). However, only the SBPF3 group revealed a notable improvement at week 18 (*p* < 0.05). In addition, SBP had both linear and quadratic effects (P_Q_ < 0.05) on tibia length.

### 3.2. SBP Supplementation Increases the Serum Immune-, Inflammatory-, and Antioxidant-Related Indices of Roosters

As shown in Figure 1A–C, SBP supplementation significantly increased serum IgA and IgM levels (*p* < 0.05), but had no significant effect on IgG (*p* > 0.05), and there was no linear or quadratic effect (P_L_ > 0.05, P_Q_ > 0.05). Furthermore, the serum contents of the three immunoglobulins in the SBPF3 group were the highest compared with the other groups.

Serum interleukin levels are shown in Figure 1D–G. The addition of SBP at 0.30% and 0.45% significantly reduced the roosters’ serum IL-1β levels (*p* < 0.05). Moreover, feeding graded concentrations of SBP decreased serum IL-8 and IL-12 levels to varying degrees (*p* < 0.05), but there was no significant influence on IL-6 (*p* < 0.05). Furthermore, SBP supplementation exhibited a quadratic effect (P_Q_ < 0.05) on serum IL-6 and IL-12 levels.

Serum T-AOC levels in the SBPF1, SBPF2, and SBPF3 groups were all significantly higher than those in the CONF group (Figure 1H, *p* < 0.05). In addition, serum levels of GSH-Px and SOD in the SBPF2 and SBPF3 groups were also elevated (Figure 1I,J, *p* < 0.05), while MDA content was significantly decreased (Figure 1K, *p* < 0.05). Moreover, the results again showed a quadratic trend (P_Q_ < 0.05) on serum GSH-Px and SOD levels of roosters fed SBP-supplemented diets.

### 3.3. SBP Supplementation Improves the Intestinal Morphology of Roosters

Dietary supplementation with varying doses of SBP improved the morphological structure of the intestine to some extent (Figure 2A–J). Specifically, SBP intake increased villus length and decreased crypt depth in the small intestine (*p* < 0.05), and, accordingly, increased the ratio of villus length to crypt depth (V/C). Compared with the other groups, 4.5 g/kg was the optimal supplement concentration. The improvement effect of SBP on small intestine morphology again conformed to a quadratic trend (P_Q_ < 0.05).

### 3.4. SBP Supplementation Increases the Intestinal Tight Junction-Gene Expression of Roosters

Figure 3A–G shows the expression of tight junction proteins in the intestine. With increasing doses of added SBP, mRNA expression of ZO-1, Occludin, and Claudin-3 all showed increasing trends (*p* < 0.05), consistent with linear and quadratic effects (P_L_ < 0.05, P_Q_ < 0.05). The expression of ZO-1 protein in the small intestine was also detected by immunofluorescence, and the results were consistent with the mRNA expression trend. The effect of SBP on the expression of tight junction proteins in the small intestine again conformed to a quadratic trend (P_Q_ < 0.05), first increasing and then decreasing with gradually increasing doses.

### 3.5. SBP Supplementation Prevents the Intestinal Cell Apoptosis of Roosters

We used the TUNEL assay (Figure 4A–D) to detect intestinal cell apoptosis, and the results indicated that exogenous SBP substantially decreased the number of positive cells in the duodenum, jejunum, and ileum (*p* < 0.05) in a dose-dependent manner (P_L_ < 0.05).

### 3.6. SBP Supplementation Enhances the Intestinal Immunity and Antioxidant Capacity of Roosters

SBP supplementation at gradually increasing concentrations reduced the expression of IFN-α and IFN-γ mRNAs in the duodenum, jejunum, and ileum, to varying degrees (Figure 5A,B, *p* < 0.05). Moreover, expression of TGF-β1 in the small intestine of roosters in the SBPF2, SBPF3, and SBPF4 groups was substantially upregulated (Figure 5C, *p* < 0.05). SBP supplementation had both dose-dependent and quadratic effects on intestinal IFN expression (P_L_ < 0.05, P_Q_ < 0.05).

SBP supplementation had a significant effect on intestinal antioxidant capacity (Figure 5D–G). mRNA expression of SOD, GSR, GST, and GPx was considerably elevated in the small intestine of roosters treated with four different doses of SBP (*p* < 0.05). SBP supplementation, again, had a quadratic influence on the expression of SOD, GSR, and GPx (P_Q_ < 0.05), but a linear impact on GST expression (P_L_ < 0.05).

### 3.7. SBP Supplementation Enhances the Reproductive Performance of Roosters

The effects of SBP supplementation on immune and reproductive-organ indices are shown in Figure 6A–D. Compared with the CONF group, spleen weight, spleen index, testicular weight, and testicular index of the four experimental groups were all dramatically increased (*p* < 0.05). Moreover, SBP supplementation improved the spleen weight and spleen index of roosters in a quadratic manner (P_Q_ < 0.05).

Indicators of rooster semen quality are shown in Figure 6E–I. SBP supplementation significantly increased ejaculation volume (*p* < 0.05), and sperm density and sperm motility were also markedly improved (*p* < 0.05). Also, semen pH rose somewhat (i.e., became less acidic; *p* < 0.05). Moreover, there was a significant decline in sperm abnormality rate (*p* < 0.05). Observation of testicular tissue sections (Figure 6J–L) further revealed that SBP supplementation significantly increased seminiferous tubule diameter and seminiferous epithelium thickness (*p* < 0.05).

### 3.8. SBP Supplementation Improves the Cecal Microbiome Composition of Roosters

Figure 7A–F shows the proportion of the top 30 microbial species in each group at the phylum level. SBP supplementation increased the relative abundance of Verrucomicrobiota in the cecal microbiota. In addition, the alpha diversity indices (Pielou-E and Shannon) in the SBPF3 group were significantly higher than those in the CONF group (Figure 5B,C, *p* < 0.05). Moreover, the NMDS and PCoA results indicated that the microbial structure of the roosters’ ceca changed, to some extent, after the addition of exogenous SBP, and there was a detectable difference between the SBPF groups and the CONF group (Figure 5D,E). Furthermore, phenotype prediction of microbiome samples by BugBase revealed that SBP supplementation increased the relative abundance of anaerobic bacteria and bacteria with biofilm-forming phenotypes, and decreased the relative abundance of Gram-negative bacteria and potentially pathogenic bacteria (Figure 7G–J).

## 4. Discussion

In this study, the addition of different levels of SBP to the diet significantly improved various growth performance indicators of roosters. These results are consistent with previous studies of bioactive peptides in broilers. With SBP concentration increased to 5 g/kg, the feed conversion rate of broilers was significantly increased [12]. In addition, adding 200 and 250 mg/kg Canola Bioactive Peptides to the diet increased BW and decreased the F/G of broilers [16]. The improvement in growth performance was mainly attributed to the nutritional functions of small-molecule active peptides. SBP is the final product of protein digestion and can be directly absorbed and utilized by the gastrointestinal tract; it can also promote the absorption of amino acids and alleviate absorption competition among them [9,17]. Previous studies have shown that appropriate application of SBP can reduce the energy and protein levels of feed without affecting the normal production of poultry, thus saving feed costs.

IgA, IgM, and IgG are the three main animal immunoglobulins, which are part of humoral immunity and have important functions as bactericides, as well as in immune regulation and preventing pathogen invasion [18,19,20]. As important lymphokines, interleukins participate in the activation and regulation of immune cells in the inflammatory response of the body, which is crucial for maintaining homeostasis [21,22]. GSH-Px and SOD are two important free-radical scavenging antioxidant enzymes, while MDA is the end product of lipid oxidation reactions. T-AOC can reflect the total antioxidant level composed of various antioxidant substances and enzymes in the body [23]. These aforementioned factors play crucial roles in the balance of immune inflammation, oxidation, and antioxidants. The present study demonstrated that SBP could increase the levels of serum immunoglobulin and antioxidant factors and reduce the levels of serum inflammatory factors in roosters. These results suggest that the three factors were better balanced, and the improvement in immune and antioxidant levels ensured a reduction in inflammation level. Similarly, dietary SBP could alleviate the increase in serum proinflammatory cytokines caused by coccidioidal challenge. Therefore, additives in poultry feed should primarily aim to enhance the levels of immunoglobulins and antioxidant enzymes. It has also been reported that active peptides can promote the development of the spleen and the response of immune cells to play an immunomodulatory role [24].

Previous reports showed that addition of SBP to the feed could improve the apparent digestibility of ileal nutrients and energy utilization of broilers [12]. Improvement of intestinal digestibility is closely related to the integrity of intestinal morphology and structure. In the present study, supplementation with SBP significantly increased villus height in the duodenum, jejunum, and ileum of roosters and decreased the depth of the crypts, thereby reducing the V/C ratio. Fermented soybean meal has been shown to have the same effect [25]. These results verified the bioactivity of soybean peptides in stimulating intestinal development and ensuring intestinal structural and functional integrity.

As the largest immune organ of poultry, the gut has an important barrier function. Tight junction proteins are an important part of the intestinal mechanical barrier, and their expression levels can directly affect the integrity of the barrier, thereby affecting various intestinal functions [26,27]. In addition, other immune and antioxidant factors also have an integral impact on gut health. Our experimental results suggested that SBP intake increased the expression of tight junction proteins at both mRNA and protein levels, promoted expression of intestinal anti-inflammatory factors, and inhibited expression of pro-inflammatory factors. Further, antioxidant levels in the intestine were also improved. This was consistent with the findings that antimicrobial peptides increased intestinal SOD activity and decreased MDA levels [28]. Malondialdehyde (MDA) is produced through lipid peroxidation of cell membranes, resulting in cell damage. As the final product, MDA content indirectly reflects the degree of lipid peroxidation and can be measured accordingly [29]. It is worth mentioning that the increased levels of immunity and antioxidants we observed may be important reasons for the significant decrease in the level of intestinal apoptosis.

Previous research has demonstrated that SBP supplementation improves the growth and production performance of layers and broilers, but the effect of SBP on male reproductive performance has remained unclear. The present results showed that SBP promoted development of the testis and improved semen quality in roosters [13]. We hypothesize that the immune/antioxidant activity of active peptides and the function of promoting nutrient absorption and utilization are important reasons for the improvement in the reproductive performance of roosters. Some peptides can bind to minerals to improve their absorption and utilization [30]. Protein and trace elements are essential for the development of reproductive organs and spermatozoa.

Intestinal microbes not only participate in the digestion, absorption, and metabolism of nutrients in the body but also play an important role in the intestinal immune barrier. Dysregulation of the gut microbiome can lead to various intestinal diseases [31]. In addition, obesity and diabetes are also closely related to the disturbance of the gut microbiome [32]. Here, we analyzed the effects of SBP supplementation on gut microbiota in roosters by 16S rRNA gene sequencing. Our analysis (Figure 7A,F–J) revealed that the relative abundance of Verrucomicrobiota in the experimental group was improved by the treatment. Akkermansia muciniphila, a member of Verrucomicrobiota, not only protects the integrity of intestinal epithelial cells and mucus layer but also plays an important anti-inflammatory role in inflammatory response [33,34]. Two alpha diversity indices (Pielou-E and Shannon) in the SBPF3 group were higher than those in the CONF group, indicating that SBP promoted the diversity and evenness of intestinal microorganisms. Moreover, beta diversity analysis also revealed differences in cecal microbial composition between the experimental and control groups. Notably, the number of bacteria with beneficial phenotypes was higher in the experimental group compared with the control group, and the relative abundance of potentially pathogenic bacteria decreased.

## 5. Conclusions

In conclusion, dietary supplementation with different levels of SBP can improve the growth performance, reproductive performance, immune function, and antioxidant capacity of roosters. In addition, SBP supplementation promoted intestinal structural integrity and increased cecal microbial diversity. In these subjects and experimental designs, supplementing the basal diet with 0.45% SBP yielded the most significant improvement in the aforementioned indices in roosters. Our results can be used as a reference for further improvement in chicken breeding and management.

## Figures and Tables

**Figure 1 animals-14-01954-f001:**
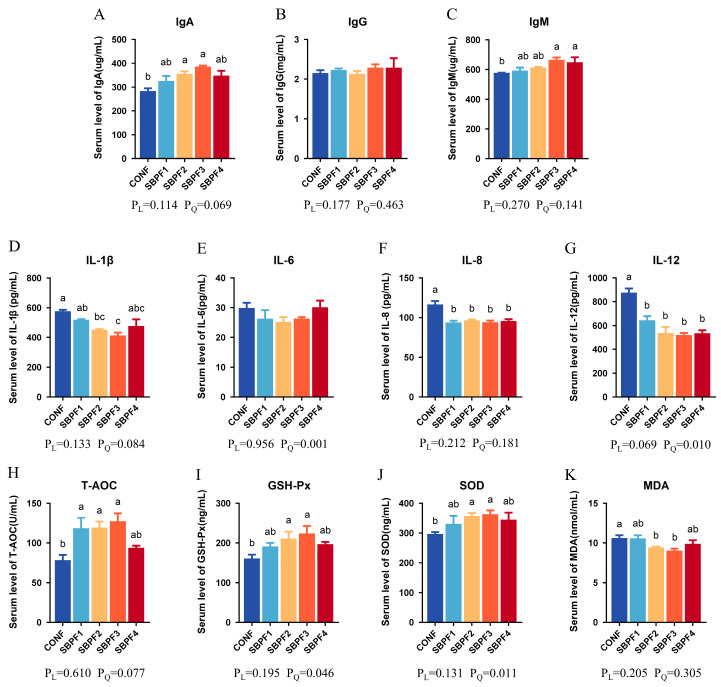
Feed supplementation with different concentrations of SBP affects the serum factors of roosters at 30 weeks of age. SBP supplementation levels are 0%, 0.15%, 0.30%, 0.45%, and 0.60% for CONF, SBPF1, SBPF2, SBPF3, and SBPF4, respectively. (**A**–**C**) Serum levels of IgA, IgG and IgM. (**D**–**G**) Serum levels of IL-1β, IL-6, IL-8 and IL-12. (**H**–**K**) Serum levels of T-AOC, GSH-Px, SOD, and MDA. Bars having different superscript letters differed significantly (*p* < 0.05).

**Figure 2 animals-14-01954-f002:**
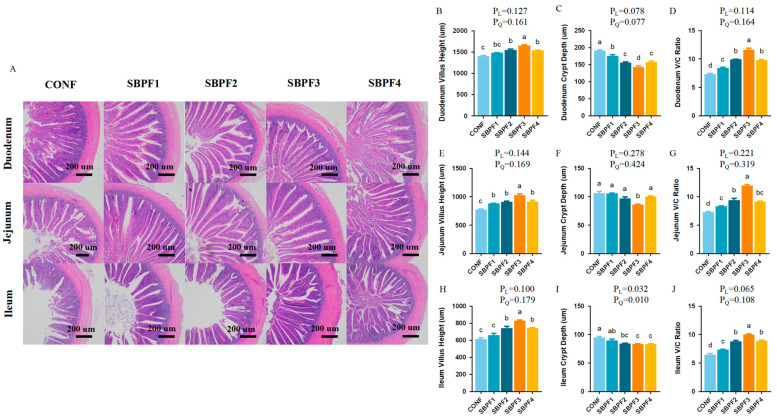
Feed supplementation with different concentrations of SBP improves the intestinal morphology of roosters at 30 weeks of age. SBP supplementation levels are 0%, 0.15%, 0.30%, 0.45%, and 0.60% for CONF, SBPF1, SBPF2, SBPF3, and SBPF4, respectively. (**A**) Villus length and crypt depth were observed in sections of different intestinal segments (40×). (**B**–**D**) Villus height, crypt depth, and V/C ratio of the duodenum. (**E**–**G**) Villus height, crypt depth, and V/C ratio of the jejunum. (**H**–**J**) Villus height, crypt depth, and V/C ratio of the ileum. Bars having different superscript letters differed significantly (*p* < 0.05).

**Figure 3 animals-14-01954-f003:**
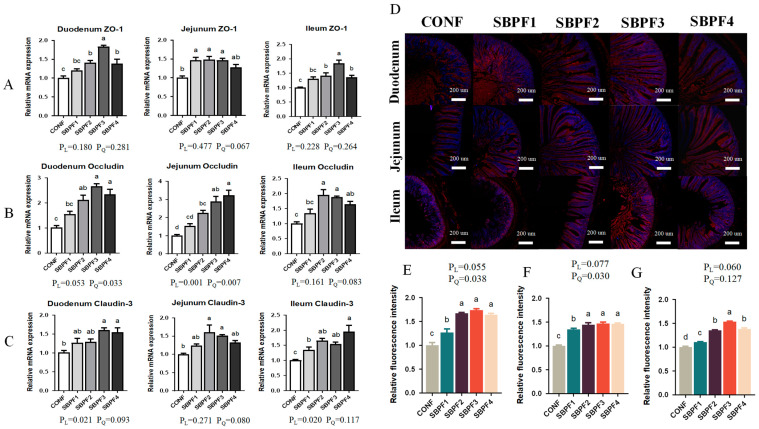
Feed supplementation with different concentrations of SBP increases the intestinal tight junction-gene expression of roosters at 30 weeks of age. (**A**–**C**) mRNA expression of ZO-1, Occludin, and Claudin-3 in the small intestinal segments (duodenum, jejunum and ileum). (**D**) Representative images of ZO-1 protein immunofluorescence. (**E**–**G**) Relative fluorescence intensity of ZO-1 protein in the small intestinal segments (duodenum, jejunum and ileum). Bars having different superscript letters differed significantly (*p* < 0.05).

**Figure 4 animals-14-01954-f004:**
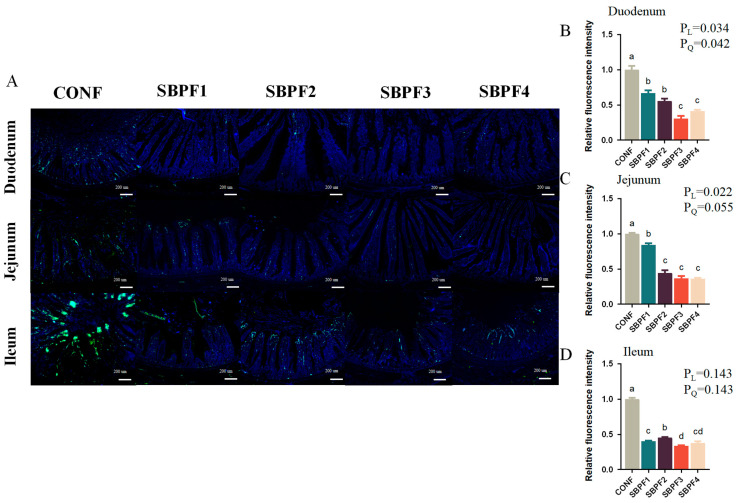
Feed supplementation with different concentrations of SBP prevents the intestinal cell apoptosis of roosters at 30 weeks of age. (**A**) Representative images of TUNEL analysis in the small intestinal segments (duodenum, jejunum and ileum). (**B**–**D**) Relative fluorescence intensity of fluorescently labeled DNA fragments in the small intestinal segments (duodenum, jejunum and ileum). Bars having different superscript letters differed significantly (*p* < 0.05).

**Figure 5 animals-14-01954-f005:**
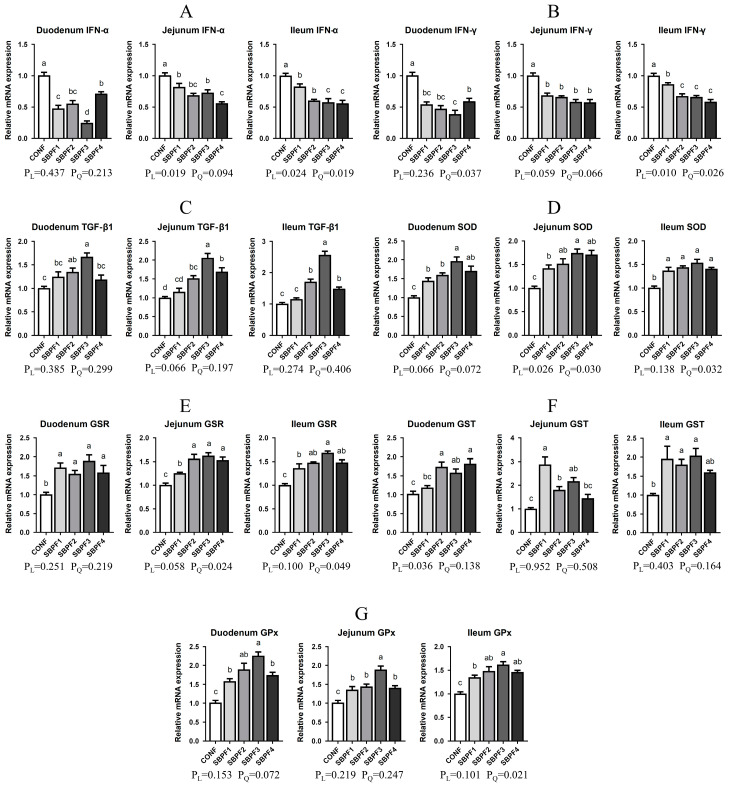
Feed supplementation with different concentrations of SBP regulates the expression of intestinal immune and antioxidant factors of roosters at 30 weeks of age. SBP supplementation levels are 0%, 0.15%, 0.30%, 0.45%, and 0.60% for CONF, SBPF1, SBPF2, SBPF3, and SBPF4, respectively. (**A**–**G**) mRNA expression of IFN-α, IFN-γ, TGF-β1, SOD, GST, GSR, and GPx in the different intestinal segments (duodenum, jejunum, and ileum). Data represent the mean ± SEM (*n* = 3 independent samples). Bars having different superscript letters differed significantly (*p* < 0.05).

**Figure 6 animals-14-01954-f006:**
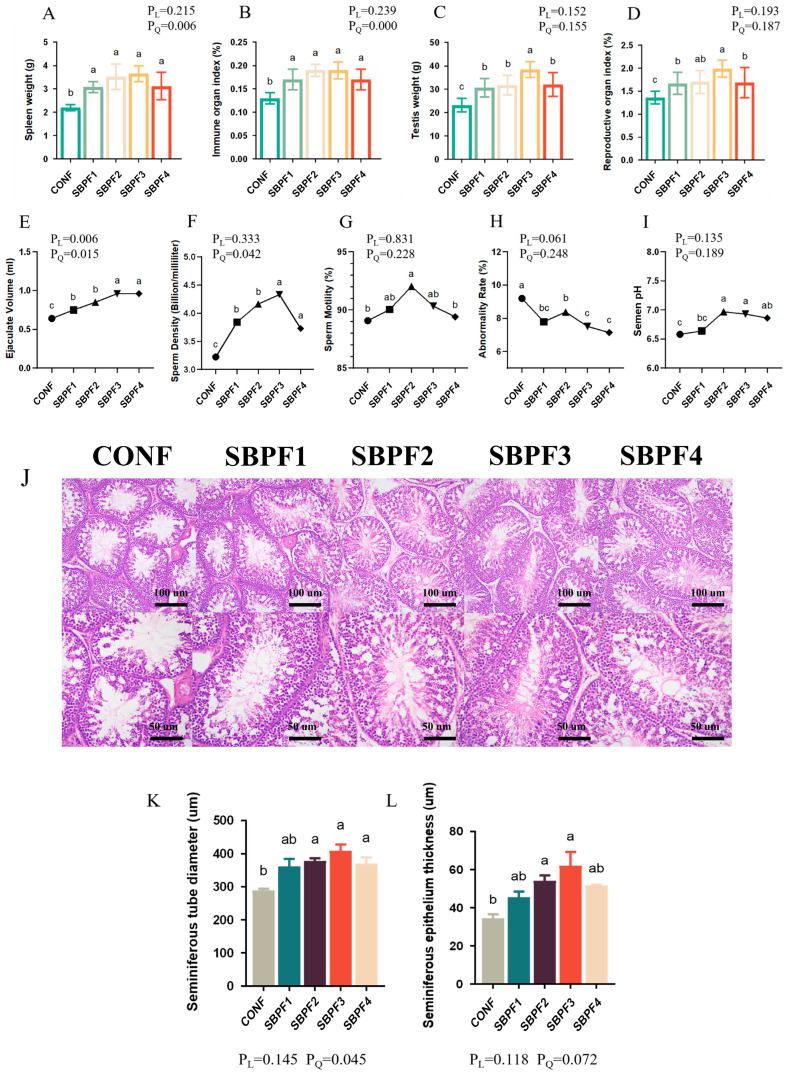
Feed supplementation with different concentrations of SBP improves the reproductive performance of roosters at 30 weeks of age. SBP supplementation levels are 0%, 0.15%, 0.30%, 0.45%, and 0.60% for CONF, SBPF1, SBPF2, SBPF3, and SBPF4, respectively. (**A**–**D**) Spleen weight, immune organ index, testis weight, and reproductive-organ index. (**E**–**I**) Ejaculate volume, sperm density, sperm motility, abnormal sperm rate and semen pH. (**J**) Images of HE-stained sections of testicular tissue. (**K**,**L**) Seminiferous tubule diameter and seminiferous epithelium thickness of the testis. In all histograms, data represent the mean ± SEM (*n* = 3 independent samples). Bars having different superscript letters differ significantly (*p* < 0.05).

**Figure 7 animals-14-01954-f007:**
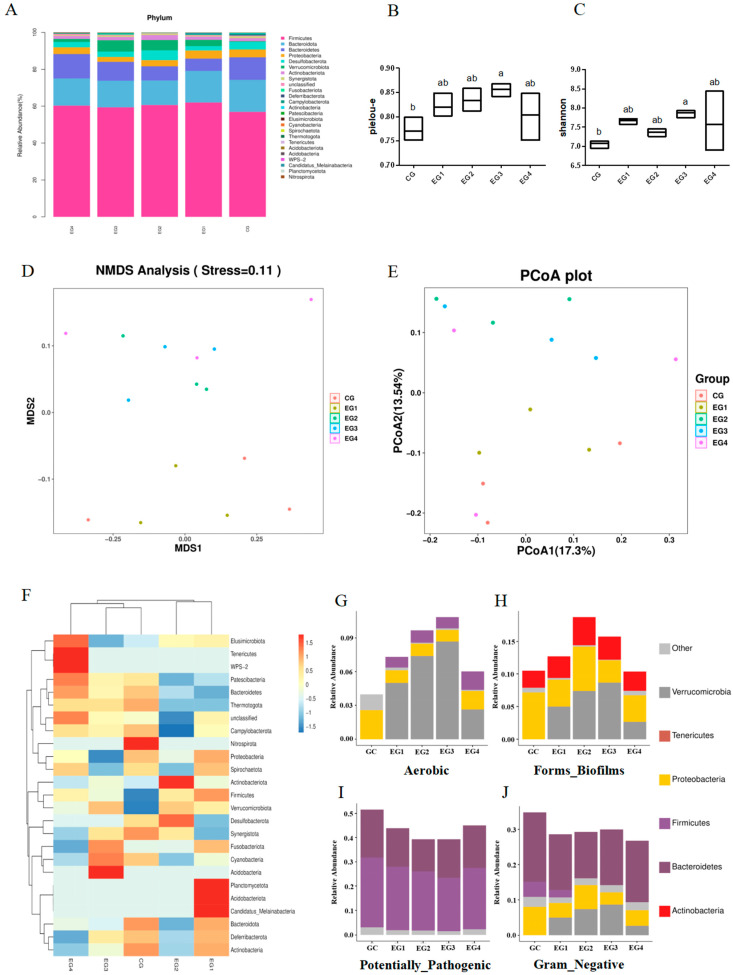
Feed supplementation with different concentrations of SBP affects the cecal microbial diversity of roosters at 30 weeks of age. (**A**) Stacked bar plot of relative abundance of top 30 microbial species at the phylum level. (**B**,**C**) Alpha diversity indices (Pielou-E and Shannon) for each group. (**D**) NMDS analysis: the closer the samples are, the more similar the microbial composition and the smaller the differences between the samples. (**E**) Principal coordinates analysis: the closer the samples are, the more similar the microbial composition and the smaller the differences between the samples. (**F**) Heat map of relative abundance of top 30 microbial species at the phylum level. (**G**) Relative abundance of phylum-level species with aerobic phenotypes. (**H**) Relative abundance of phylum-level species with biofilm-formation phenotypes. (**I**) Relative abundance of phylum-level species with potentially pathogenic phenotypes. (**J**) Relative abundance of phylum-level species with Gram-negative phenotypes. Bars having different superscript letters differ significantly (*p* < 0.05).

**Table 1 animals-14-01954-t001:** Effects of SBP on rooster growth performance.

Parameters	Age	CONF	SBPF1	SBPF2	SBPF3	SBPF4	SEM	P_L_	P_Q_
BW (g)	0	33.40	33.74	34.34	33.84	34.18	1.83	0.150	0.200
	6	370.40 ^b^	385.40 ^a^	402.40 ^a^	413.00 ^a^	405.00 ^a^	16.27	0.042	0.036
	18	1419.00 ^c^	1495.00 ^bc^	1570.00 ^ab^	1653.00 ^a^	1559.00 ^ab^	57.44	0.111	0.118
	30	1648.00 ^c^	1726.00 ^bc^	1807.00 ^ab^	1896.00 ^a^	1798.00 ^ab^	48.72	0.106	0.120
Tibia length (mm)	0	27.45	26.08	27.06	26.83	27.85	0.84	0.544	0.357
	6	66.77 ^b^	68.55 ^ab^	69.17 ^ab^	70.02 ^a^	70.78 ^a^	2.19	0.004	0.017
	18	96.49 ^b^	100.03 ^ab^	99.04 ^ab^	100.19 ^a^	98.52 ^ab^	2.39	0.450	0.250
ADG (g/d)	1–6	8.04 ^c^	8.37 ^bc^	8.73 ^ab^	9.05 ^a^	8.82 ^a^	0.38	0.057	0.043
	7–18	12.45 ^d^	13.24 ^c^	13.93 ^b^	14.73 ^a^	13.75 ^bc^	0.77	0.132	0.121
	19–30	2.75	2.77	2.81	2.86	2.85	0.10	0.013	0.075
ADFI (g/d/bird)	1–6	11.79	12.09	12.1	12.11	12.17	0.30	0.087	0.138
	7–18	57.17	57.74	58.00	57.98	57.72	0.63	0.252	0.001
	19–30	90.33	90.84	90.9	90.91	90.76	0.53	0.275	0.049
F/G (g/g)	1–6	1.47 ^a^	1.45 ^ab^	1.38 ^ab^	1.34 ^b^	1.38 ^ab^	0.06	0.070	0.145
	7–18	4.60 ^a^	4.36 ^ab^	4.17 ^bc^	3.94 ^c^	4.20 ^b^	0.23	0.113	0.104

^abcd^ Mean values in the same row having different superscript letters differed significantly (*p* < 0.05). P_L_: linear *p* value, P_Q_: quadratic *p* value. Abbreviations: BW, body weight; ADG, average daily gain; ADFI, average daily feed intake; F/G, feed-to-gain ratio.

## Data Availability

Data are contained within the article.

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
