# Peer review of "Soybean Bioactive Peptide Supplementation Affects the Intestinal Immune Antioxidant Function, Microbial Diversity, and Reproductive Organ Development in Roosters"

_animals, 2024, doi:10.3390/ani14131954_

Round 1
Reviewer 1 Report
Comments and Suggestions for Authors
The article presents very robust data and is an important contribution to the field of study. The suggestions provided aim to further enhance the quality of the work. It was a pleasure to review this article.
My considerations are as follows:
1. I recommend bringing the information from lines 94 and 95 to the beginning of the paragraph to improve clarity and flow of the text.
2. In the methodology section, it is essential to include the location and site where the experimental phase of this study took place. This will enrich the description and provide a more complete context to the research.
3. It is important to include information about the weight of one-day-old chicks. Furthermore, it should be considered whether the term "roosters" is appropriate for one-day-old chicks; the correct term may be "one-day-old chicks".
4. Reorganize the description of the experimental design: what is distributed are the treatments, not the animals. Therefore, the correct sentence would be: "The treatments were randomly distributed into 5 groups, with 5 repetitions of 30 animals each".
5. The formulation of the basal diet and the four experimental groups is important and should be included in the methodology.
6. I suggest adding an item detailing the management of the roosters during the experimental period, addressing aspects such as lighting, cleaning of feeders and drinkers.
7. Clarify the housing structure of the roosters, including details on the types of feeders and drinkers used, the size of the pens, the type of bedding adopted, the lighting and ventilation system, the presence of fans or side curtains, and the methods of collecting ambient temperature.
8. It is essential to mention all these points in the methodology to provide a complete description of the experimental environment.
9. Include in the methodology the description of tibia length measurements.
10. In the first table and figure, explain the meaning of the acronyms/abbreviations. In subsequent tables and figures, use only the acronyms/abbreviations. Correct lines 182 and 204 as necessary.
11. I also suggest adding captions to Table 1 explaining the abbreviations. Instead of "item," use "variables" or "parameters."
12. The data presentations in the form of figures are well done; however, some figures are small, making it difficult to visualize the results. I recommend increasing the size of Figure 2B and reorganizing Figures 3A, 3B, 3C, 3E, 3F, and 3G, as well as Figures 4B-D to improve visualization.
13. Correct line 284: "figure 4J-L" to "figure 6J-K".
14. Correct line 198: the T-AOC data is in Figure 1H only.
15. Correct line 258: "figure 5A-B" instead of "figure 5A-C".
16. The results are well written and easy to read, in accordance with the presented figures.
17. Regarding the discussion, each paragraph should focus on a single topic to maintain cohesion and a logical sequence according to the results.
18. Develop the results related to immunoglobulins and antioxidant enzymes better, reinforcing their potential as an efficient additive in poultry farming.
19. Also reinforce the importance of reducing MDA levels, as this is a significant result.
20. Strengthen the conclusion by highlighting the practical importance of the results.
21. Regarding references:
- Correct all references as necessary.
- Cite all authors, removing "et al."
- The year should be in bold.
- The journal name should be abbreviated and in italics (CORRECT 1, 10, 12, 16, 17, 20).
- Check reference 23.
Reviewer 2 Report
Comments and Suggestions for Authors
Soybean bioactive peptides are small protein molecules produced by in vitro enzymatic hydrolytic processes and derived from soybeans that have been shown to have antioxidant, anti-inflammatory, and immunomodulatory properties. When administered as a dietary supplement to poultry diets, soybean bioactive peptides have been found to positively impact the health and productive performance of birds. However, the current study aims to evaluate the impact of different levels of soybean bioactive peptide feed additives (0%, 0.15%, 0.30%, 0.45%, and 0.60%) on the health and reproductive performance of roosters. It is an interesting article that deals with an important vital topic and is well written. In my opinion, this article can be accepted in its current form.
Comments on the Quality of English LanguageAcceptable.
